# A scalable peptide-GPCR language for engineering multicellular communication

Sonja Billerbeck[1], James Brisbois[1], Neta Agmon[2], Miguel Jimenez [1,4], Jasmine Temple[2], Michael Shen[2], Jef D. Boeke [2] & Virginia W. Cornish[1,3]

Engineering multicellularity is one of the next breakthroughs for Synthetic Biology. A key bottleneck to building multicellular systems is the lack of a scalable signaling language with a large number of interfaces that can be used simultaneously. Here, we present a modular, scalable, intercellular signaling language in yeast based on fungal mating peptide/G-protein-coupled receptor (GPCR) pairs harnessed from nature. First, through genome-mining, we assemble 32 functional peptide-GPCR signaling interfaces with a range of dose-response characteristics. Next, we demonstrate that these interfaces can be combined into two-cell communication links, which serve as assembly units for higher-order communication topologies. Finally, we show 56 functional, two-cell links, which we use to assemble three- to six-member communication topologies and a three-member interdependent community. Importantly, our peptide-GPCR language is scalable and tunable by genetic encoding, requires minimal component engineering, and should be massively scalable by further application of our genome mining pipeline or directed evolution.

[1] Department of Chemistry, Columbia University, New York, New York 10027, USA. [2] Institute for Systems Genetics and Department of Biochemistry and Molecular Pharmacology, NYU Langone Health, 430 East 29th Street, New York 10016, USA. [3] Department of Systems Biology, Columbia University, New York, New York 10032, USA. [4] Present address: The Koch Institute for Integrative Cancer Research, Massachusetts Institute of Technology, Cambridge, Massachusetts 02139, USA. These authors contributed equally: Sonja Billerbeck, James Brisbois, Neta Agmon.  Correspondence and requests for materials should be addressed to V.W.C. (email: vc114@columbia.edu)

The step from unicellular to multicellular organisms is considered one of the major transitions in evolution[1]. Phylogenetic inference suggests that cell–cell communication, cell–cell adhesion, and differentiation constitute the key genetic traits driving this transition[2]. Accordingly, cell–cell communication plays an important role in many complex natural systems, including microbial biofilms[3,4], multi-kingdom biomes[5,6], stem cell differentiation[7], and neuronal networks[8]. In nature, communication between species or cell types relies on a large pool of both promiscuous and orthogonal communication interfaces, acting over short and long ranges. Signals range from simple ions and small organic molecules up to highly information-dense macromolecules including RNA, peptides, and proteins. This diverse pool of signals allows cells to process information precisely and robustly, enabling the emergence of properties, such as fate decisions, memory, and the development of form and function. In contrast, current approaches to engineering synthetic biological communication mostly rely on a single signaling modality— quorum sensing (QS), a cell density-based communication system used by many bacteria[9]. The discovery of bacterial QS almost 50 years ago[10] led to a paradigm shift in synthetic microbial ecology, enabling the engineering of systems with synthetic pattern formation[11], cellular computing[12,13], controlled population dynamics[14,15], and other emergent properties[16]. QS has been exported from bacteria into plants[17] and mammalian cells[18], and inspired our effort to build an extensible communication language.

The major class of QS is based on diffusible acyl-homoserine lactone (AHL) signaling molecules generated by AHL synthases and AHL receptors that function as transcription factors, regulating gene expression in response to AHL signals. Currently, only four AHL synthase/receptor pairs are available for synthetic communication, with three pairs successfully used together[19]. Scaling the QS components to make new orthogonal communication interfaces is challenged by the fact that many of the known receptors exhibit crosstalk[20,21]. While it is possible to eliminate crosstalk by receptor evolution[22], scaling the number of unique AHL ligand/receptor pairs by laboratory evolution requires the concerted engineering of AHL biosynthesis and receptor specificity.

Communication has also been engineered using autoinducer peptides (AIP)[23] and autoinducer molecules (AI-2)[24] from Gram-positive bacteria; however, scaling is also challenged by the interdependence of multiple required signaling components. Autoinducer peptides are a class of post-translationally modified peptides sensed by a membrane-bound two-component system[25]. AI-2 is a family of 2-methyl-2,3,3,4-tetra-hydroxytetrahydrofuran or furanosyl borate diester isomers, synthesized by LuxS from S-ribosylhomocysteine followed by cyclization to a range of AI-2 isoforms[26,27], and recognized by the transcriptional regulator LsrR[28]. It was shown that the response characteristics and the promoter specificity of LsrR can be engineered[29,30] and that cell–cell communication can be tuned by using various AI-2 analogues[24]. However, the complexity of signal biosynthesis and reliance on specific transporters for signal import and export[28] complicates the potential scalability of these systems in terms of available unique communication interfaces.

Recently, mammalian Notch receptors have been repurposed to engineer modular communication components for mammalian cells. Impressively, 16 distinct SynNotch receptors were engineered and pairs of two were employed together[31]; however, SynNotch receptors are contact-dependent and therefore are only suitable for short-range communication, which is conceptually different from long-range communication through diffusible signals.

Ideally, a synthetic language would consist of an easily scalable set of independent interaction channels without crosstalk. After demonstrating in our recent work on yeast biosensors that fungal mating GPCRs couple well to the conserved yeast MAP-kinase signaling cascade[32], we hypothesized that the peptide/GPCR-based mating language of fungi could be harnessed as an ideal source of modular parts for a scalable communication language.

Fungi use peptide pheromones as signals to mediate highly orthogonal, species-specific mating reactions[33]. These peptides are genetically encoded, translated by the ribosome, and the alpha-factor-like peptides, which are typified by the 13-mer *S. cerevisiae* mating pheromone alpha-factor, are secreted through the canonical secretion pathway without covalent modifications. Peptide pheromones are sensed by specific GPCRs (Ste2-like GPCRs) that initiate fungal sexual cycles[34]. Importantly, these peptide pheromones (9–14 amino acids in length) are rich in molecular information and the composition of peptide pheromone precursor genes is modular, consisting of two N-terminal signaling regions—pre and pro—that mediate precursor translocation into the endoplasmic reticulum and transiting to the Golgi, followed by repeats of the actual peptide sequence separated by protease processing sites. This modular precursor composition allows bioinformatic inference of mature peptide ligand sequences from available genomic databases. GPCRs from mammalian and fungal origin have been used on a small scale (two to three GPCRs) to engineer programmed behavior and communication[35,36] and cellular computing[37]. However, the potential of leveraging the vast number of naturally evolved mating peptide-GPCR pairs as a scalable signaling language remains untapped.

In order to challenge the inherent scalability of the fungal mating components as a synthetic signaling language, here we establish a pipeline for language component acquisition and communication assembly (Fig. 1a): We first genome-mine an array of peptide-GPCR pairs and verify GPCR functionality and peptide secretion. Next, we couple GPCR activation to peptide secretion to validate their functionality as orthogonal communication interfaces. Those interfaces are then used to assemble scalable communication topologies and eventually to establish peptide signal-based interdependence as a strategy to assemble multi-member microbial communities. Our language acquisition pipeline shows a hit rate of 71%. Out of 45 tested GPCRs, 32 are functionally expressed and activated by a peptide ligand that was correctly inferred from its genomic locus architecture. Of these, 50% are highly orthogonal, yielding 17 unique communication channels without engineering. Importantly, our set includes peptide-GPCR pairs derived from a wide range of species from the whole Ascomycete phylum. As such, we expect that many (likely hundreds or even thousands) additional orthogonal channels are available for extraction using the workflow described herein.

## Results

**Genome-mining yields a pool of functional peptide-GPCR pairs**. First, we mined a total of 45 peptide-GPCR pairs from available Ascomycete genomes (Supplementary Table 1); sequences of mature peptide ligands were taken from literature (Supplementary Table 1) or inferred from peptide precursor sequences (Supplementary Table 2). In some cases, inference of mature peptide sequences was hampered by ambiguous protease processing sites or sequence-variable peptide repeats. The GPCR's tolerance to sequence variation in its peptide ligands was evaluated by incorporating alternate peptide sequence candidates into our analysis (Supplementary Table 1 and 2). Functionality of heterologous mating GPCRs in *S. cerevisiae* requires proper

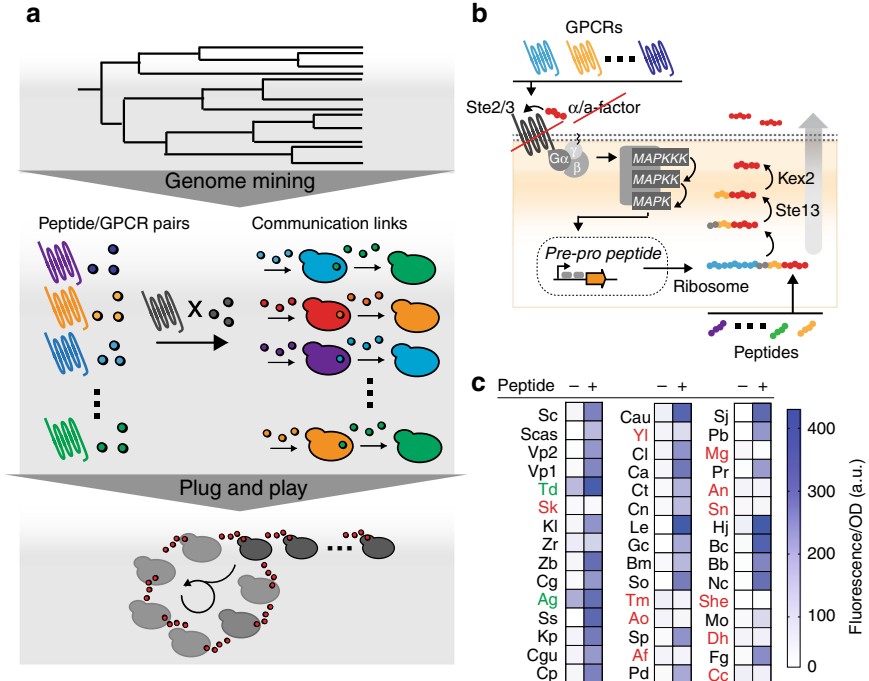

**Fig. 1** Language component acquisition: genome-mining yields a scalable pool of peptide-GPCR interfaces. **a** Pipeline for component harvest and communication assembly. Upper panel: mining of Ascomycete genomes yields a scalable pool of peptide-GPCR pairs. Middle panel: GPCR activation can be coupled to peptide secretion to establish two-cell communication links. Each cell senses an incoming peptide signal via a specific GPCR, with GPCR activation leading to secretion of an orthogonal user-chosen peptide. The secreted peptide serves as the outgoing signal sensed by the second cell. Lower panel: Scalable communication networks can be assembled in a plug-and play manner using the two-cell communication links. **b** GPCRs and peptides can be swapped by simple DNA cloning. Conservation in both GPCR signal transduction and peptide secretion enables scalable communication without any additional strain engineering. Mating GPCRs couple to the *S. cerevisiae* Gα protein (Gpa1) and signals are transduced through a MAP-kinase-mediated phosphorylation cascade. Gene activation is then mediated by the transcription factor Ste12 through binding of a pheromone response element (PRE, gray) in the promoters of mating-associated genes (e.g., *FUS1* and *FIG1*, used herein to control synthetic constructs of choice). Peptides are translated by the ribosome as pre-pro peptides. Pre-pro peptide architecture is conserved and starts with an N-terminal secretion signal (light blue), followed by Ste13 and Kex2 recognition sites (gray and yellow, respectively). Mature secreted peptides (red) are processed while trafficking through the ER and Golgi. The conserved pre-pro-peptide architecture enables the bioinformatic deorphanization of fungal GPCRs by inference of mature peptide sequences from precursor genes. **c** Most genome-mined peptide-GPCR pairs are functional in yeast. Functionality of 45 peptide-GPCR pairs was evaluated by on/off testing using 40 μM cognate peptide and fluorescence as read-out. GPCRs are organized by percent amino acid identity to the Sc.Ste2. Non-functional GPCRs (those that give a signal difference < 3 standard deviations) are highlighted in red; constitutive GPCRs are highlighted in green. GPCR nomenclature corresponds to species names (Supplementary Table 1). Experiments were performed in triplicate and full data sets with errors (standard deviation) and individual data points are given in Supplementary Figure 3

insertion into the membrane and coupling to the *S. cerevisiae* Gα subunit (Fig. 1b). Genome-mined GPCRs showed amino acid sequence identities between 17 to 68% to the *S. cerevisiae* mating GPCR Ste2 (Supplementary Table 1 and Supplementary Figure 1), but most of them showed higher conservation at specific intracellular loop motifs known to be important for Gα coupling[38,39] (Supplementary Figure 1, Supplementary Table 1). Functionality of peptide-GPCR pairs was assessed in a standardized workflow, in which codon-optimized GPCR genes were expressed in *S. cerevisiae* and tested for a positive response to synthetic peptide ligands using a *FUS1* promoter inducible red fluorescent protein (yEmRFP[40]) signal as a read-out. The simple chemistry of the peptide synthesis facilitated GPCR characterization, as any short peptide sequence is readily commercially available. GPCRs were expressed from the *TDH3* promoter using a low-copy plasmid. We engineered a read-out strain for our fluorescence assay by deleting both endogenous mating GPCR genes (*STE2* and *STE3*), all pheromone genes (*MFA1/2* and *MFALPHA1/MFALPHA2*), *BAR1* and *SST2* to improve pheromone sensitivity, and *FAR1* to avoid growth arrest (Supplementary Table 4). We constructed the read-out strain in both mating-type genetic backgrounds. Although we used the

*MAT***a**-type for language characterization herein, we confirmed language functionality in the *MAT*α-type using a subset of GPCRs (Supplementary Figure 2).

Remarkably, 32 out of 45 tested GPCRs (71%) gave a strong fluorescence signal in response to their inferred synthetic peptide ligand (ligand candidate #1, Supplementary Tables 1 and 2) (Fig. 1c, Supplementary Figure 3a). Two GPCRs were constitutively active and showed fluorescence levels > threefold above the basal levels of the other GPCRs in the absence of peptide, but showed an increase in activation in the presence of peptide (Fig. 1c, Supplementary Figure 3b). Eleven GPCRs did not respond to the initially inferred peptide ligand candidates (Fig. 1c, Supplementary Figure 3c). One of these 11 GPCRs (She.Ste2) could be activated when using an alternate near-cognate peptide ligand candidate (in this specific case the near-cognate candidate has two additional N-terminal residues), indicating that we had initially inferred the wrong peptide sequence (Supplementary Figure 3d).

**Peptide-GPCR pairs show a range of response characteristics.** After initial on/off screening, we measured dose-response curves

for all 32 functional GPCRs and extracted parameters crucial for establishing communication: sensitivity of GPCRs ($EC_{50}$), basal and maximal activation (fold-change activation), dynamic range (Hill coefficient), orthogonality, reversibility of signaling, and population response behavior (Fig. 2a–c, Supplementary Figure 4, Supplementary Table 5). Sensitivity of the GPCRs for their cognate ligand gave an $EC_{50}$ range of ~ 1–10⁴ nM, with the natural *S. cerevisiae* Ste2 exhibiting the highest sensitivity of 1.25 nM. This is comparable with the sensitivity of available QS systems[19]. Functional GPCRs displayed between 1.3- and 17-fold activation. While this range is on average a bit lower than the available QS systems[19], fold activation is comparable to other engineered GPCR-based signaling systems in yeast and mammalian cells[41,42]. Response behaviors ranged from a graded response (analog) with a wide dynamic range to switch-like (digital) behavior with a very narrow dynamic range. When we characterized dose responses at the single-cell level, we observed a subset of non-responding cells, likely due to plasmid copy number noise (Supplementary Figure 5a–c). Genomic integration of the GPCRs abolished this non-responding sub-population (Supplementary Figure 5d–f). Importantly, GPCR signaling could be deactivated and reactivated several times with either no or minimal lengthening of response time (Supplementary Figure 6). Pairs of GPCRs could also be co-expressed in a single cell in order to allow for processing of two separate signals by a single cell (Supplementary Figure 7).

**Many fungal peptide-GPCR pairs are naturally orthogonal.** Next, we assessed pairwise orthogonality for a subset of 30

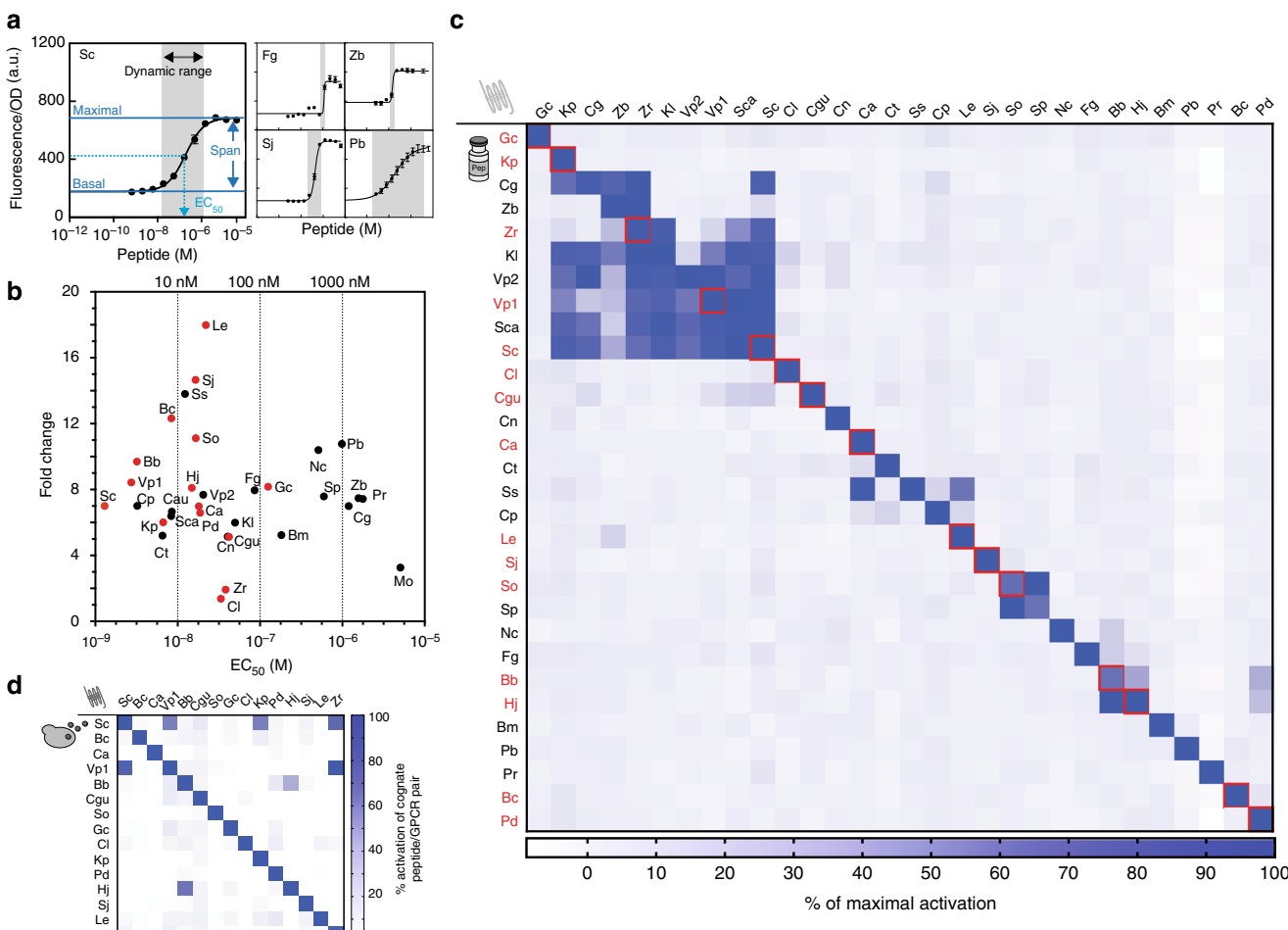

**Fig. 2** Peptide-GPCR pairs exhibit tunable response characteristics, are naturally orthogonal, and peptides are functionally secreted. **a** Experimental framework for GPCR characterization. Performance of each peptide-GPCR pair was evaluated by recording its dose-response to synthetic cognate peptides, using fluorescence as a read-out. Parameter values for basal and maximal activation, fold change, $EC_{50}$, dynamic range (given through Hill slope) were extracted by fitting each curve to a four-parameter non-linear regression model using PRISM GraphPad. Experiments were done in triplicates and errors represent the SD. Dose-response curves of GPCRs (Sc.Ste2, Fg.Ste2, Zb.Ste2, Sj.Ste2, Pb.Ste2) with different response behaviors are featured. **b** The GPCRs cover a wide range of response parameters. The $EC_{50}$ values of peptide-GPCR pairs are plotted against fold change in activation. Experiments were done in triplicate and parameter errors can be found in Supplementary Table 5. **c** GPCRs are naturally orthogonal across non-cognate synthetic peptide ligands. A 30 × 30 orthogonality matrix was generated by testing the response of 30 GPCRs across all 30 peptide ligands. The test concentration was set at 10 μM of a given peptide ligand. The fluorescence signal for maximum activation of each GPCR (not necessarily its cognate ligand) was set to 100% activation and the threshold for categorizing cross-activation was set to be ≥ 15% activation of a given GPCR by a non-cognate ligand. Experiments were performed in triplicate. GPCRs are organized according to a phylogenetic tree of the protein sequences. **d** Orthogonality of peptide-GPCR pairs when peptides are secreted. The 15 best performing pairs (marked in red in panels **a–c**) were chosen for secretion. Experiments were performed by combinatorial co-culturing of strains constitutively secreting one of the indicated peptides and strains expressing one of the indicated GPCRs using GPCR-controlled fluorescent as read-out. Experiments were performed in triplicate and results represent the mean

peptide-GPCR interfaces by exposing each GPCR to all non-cognate peptide ligands. The test concentration for assessing pair orthogonality was set at 10 μM of a given peptide ligand, and the threshold for categorizing cross-activation was set to be ≥ 15% activation of a given GPCR by a non-cognate ligand (maximum activation of each GPCR at the same concentration of the cognate ligand was set to 100% activation). As the chosen test concentration of 10 μM is three orders of magnitude higher than typically achieved by peptide secretion (1–10 nM), we rationalized that it would be a stringent selection criterion to yield peptide-GPCR pairs that would be fully orthogonal within our language. Typical values of cross-activation were between 16 and 100%. The GPCRs showed a remarkable level of natural orthogonality (Fig. 2c). In total, 14 out of 30 GPCRs were exquisitely orthogonal and only activated by their cognate peptide ligand. Five GPCRs were activated by only one additional non-cognate peptide, and 11 GPCRs were activated by several non-cognate ligands. From these results, manual curation yielded a set of 17 unique peptide-GPCR interfaces within our design constraints that can be used together in our language (17 receptors each orthogonal to all 16 non-cognate ligands) (Supplementary Figure 8).

**GPCR response characteristics are tunable by ligand recoding.** Next, we wanted to validate the robustness of our ability to infer a GPCR's peptide ligand. Thus, we recorded dose-response curves for a subset of 19 GPCRs to possible alternative near-cognate peptide ligand candidates. Fourteen out of the 19 GPCRs were also activated by these near-cognate peptides (Supplementary Figure 9), suggesting that the employed bioinformatic ligand inference strategy did not require precise interpretation of the exact precursor processing. In fact, near-cognate ligands could be harnessed to induce significant changes in $EC_{50}$, fold activation, and dynamic range for most peptide-GPCR pairs (Supplementary Figure 10). For example, the So.Ste2 changed its response characteristics from gradual to switch-like when three additional residues were included at the N-terminus of its peptide. The degree and nature of changes were unique to each GPCR/peptide pair (Supplementary Figure 10). We explored this feature further by alanine scanning the peptide ligand of the Ca.Ste2. These simple one-residue exchanges elicited shifts in $EC_{50}$ and fold change (Supplementary Figure 11). We further extended this to several promiscuous GPCRs and their cross-activating non-cognate ligands (Supplementary Figure 12). While some GPCRs retained stable response parameters across a variety of peptide ligands, most GPCRs' response parameters could be modulated when exposed to these peptide variants. Combined, these results imply the exciting opportunity to tune the response characteristics of a given GPCR by simply recoding the peptide ligand instead of engineering the receptor itself—a feature that can be exploited in future efforts.

**Peptides are functionally secreted.** After assessing peptide-GPCR functionality with synthetic peptides, we tested whether the peptides could be functionally secreted. The posibility of peptide secretion from *S. cerevisiae* through its conserved *sec* pathway has been shown before[43], but the feasibility across a wide sequence space was unclear. The amino acid sequences of 15 peptides were cloned into a peptide secretion vector, designed based on the alpha-factor pre-pro-peptide architecture (Supplementary Figure 13, Supplementary Table 6). The 15 peptides were chosen based on the favorable dose–response characteristics (low $EC_{50}$ and high fold change) of the corresponding peptide-GPCR pairs.

To test for peptide secretion, we employed the appropriate GPCR/fluorescent-read-out strains as peptide sensors in a liquid assay as well as a fluorescent halo assay. All peptides could be secreted from *S. cerevisiae* (Fig. 2d, Supplementary Figure 14 and 15) but the amount of peptide secretion was dependent on the peptide sequence (Supplementary Figure 14 and 15). Combinatorial co-culturing of secreting and sensing strains validated that peptide-GPCR pair orthogonality was retained when peptides were secreted (Fig. 2d).

**Two-cell links serve as minimal signaling units.** Next, we established functional communication by coupling GPCR signaling to peptide secretion. We conceptualized our language to be built from two-cell links as the minimal signaling units that can be easily characterized and assembled into higher-order communication topologies (Fig. 3a). In brief, in a *c1-c2* two-cell link, cell *c1* senses peptide *p1* by expressing GPCR *g1*. GPCR *g1* activation leads to secretion of peptide *p2* from cell *c1*, sensed by cell *c2* through GPCR *g2*. GPCR *g2* signaling is coupled to a fluorescent read-out. Dose-dependent transfer of information through each link can be assessed by exposing cell *c1* to an increasing dose of synthetic peptide *p1* and measuring an increase in fluorescence in cell *c2*. In this manner, each two-cell link can be characterized by a signal transfer function (*p1* dose to *c2* response) making it easy to identify optimal links for a given topology. In order to test the assembly of functional two-cell links, we chose eight fully orthogonal peptide-GPCR pairs and characterized the complete combinatorial set of 56 possible links (all possible non-cognate combinations; Fig. 3a, b, Supplementary Figure 16 and 17). In all 56 cases, activation of the *g1* GPCR resulted in a graded, *p1* concentration-dependent fluorescence signal in *c2*.

**Two-cell links can be used to build communication topologies.** Next, we tested if our language could be used to link multiple yeast strains and build synthetic multicellular communities. The functional capabilities of single engineered organisms are limited by their capacity for genetic modification. Multi-membered microbial consortia, engineered to cooperate and distribute tasks, show promise to unlock this constraint in engineering complex behavior. For example, we envision engineering sense-response consortia composed of yeast that sense a trigger, e.g., a pathogen[32], and yeast that respond, e.g., by killing the pathogen through secretion of an antimicrobial[44]. Further, consortia have shown distinct advantages for metabolic engineering, such as distribution of metabolic burden, as well as parallelized, modular optimization, and implementation[45,46]. Those consortia have applications in degrading complex biopolymers like lignin, cellulose[47], or plastic[48].

First, we combined the established two-cell communication links into a scalable paracrine ring topology. A ring is a network topology in which each cell *cx* connects to exactly two other cells (*cx*-1 and *cx* + 1), forming a single continuous signal flow. The ring topology can be efficiently scaled by adding additional links. Failure of one of the links in the ring leads to complete interruption of information flow, allowing simultaneous monitoring of the functionality and continued presence of all ring members. We combined the two-cell links into rings of increasing size, from two to six members (Fig. 3c, **topologies 1–6**). Information flow was started by cell *c1* constitutively secreting the peptide sensed by cell *c2* through GPCR *g2*. Peptide sensing in cell *c2* was coupled to secretion of peptide *p3* sensed by cell *c3* through GPCR *g3*. In this manner, peptide signals were transmitted around the ring. Our N-member ring is closed by cell *cN* secreting the peptide sensed by cell *c1* through GPCR *g1*. *c1* reports on ring closure by a GPCR-coupled fluorescence

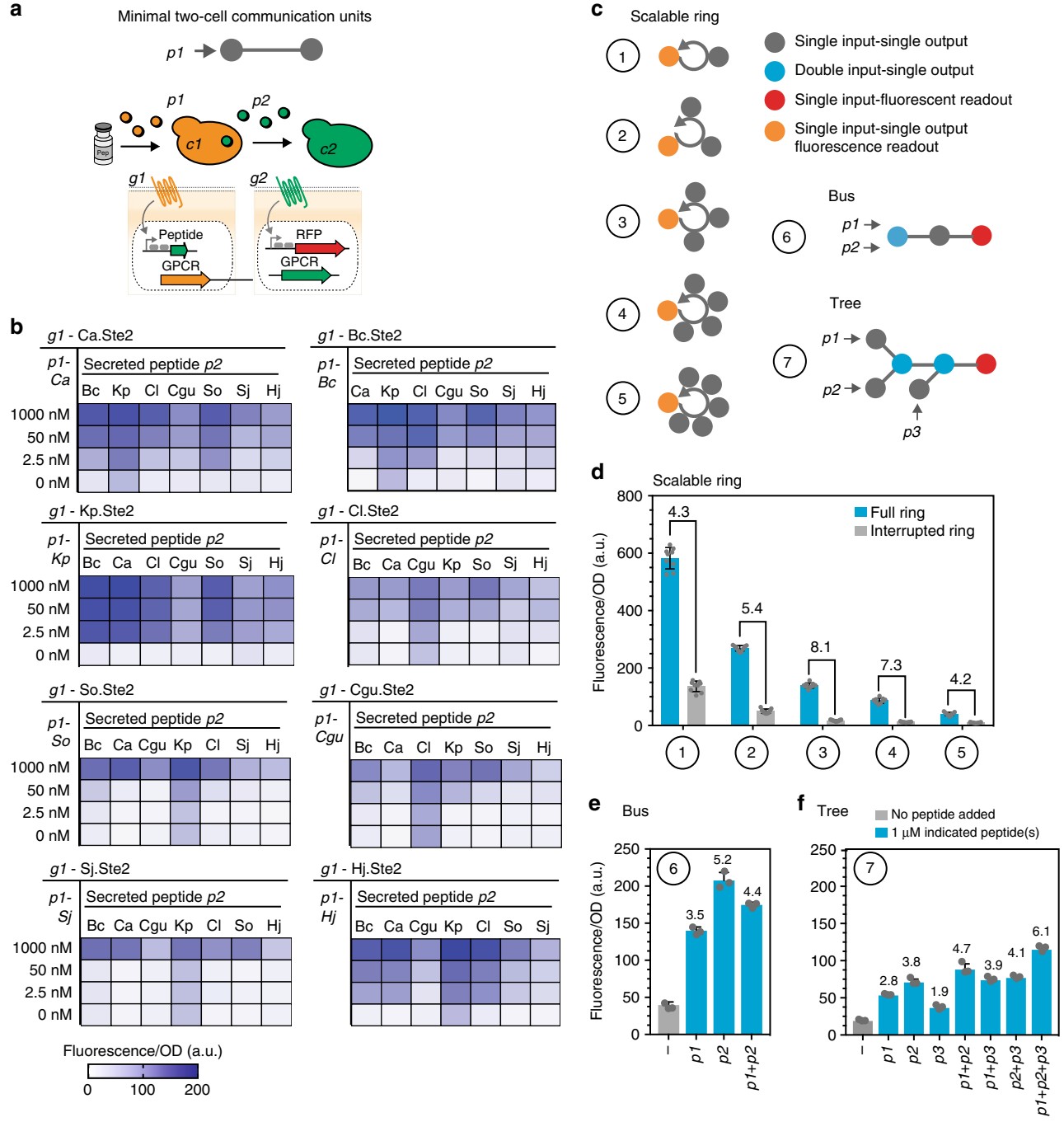

**Fig. 3** Synthetic microbial communication: Two-cell communication links yield various communication topologies. **a** Illustration of minimal two-cell links. Cell 1 (*c1*) senses synthetic peptide through GPCR 1 (*g1*). Activation of *g1* leads to secretion of peptide 2 (*p2*). *p2* is sensed by cell 2 (*c2*) through GPCR 2 (*g2*). *g2* activation is coupled to a fluorescent read-out. Signal transmission from *c1* to *c2* is assessed by recording transfer-functions using co-cultures of *c1* and *c2* and increasing amounts of *p1*. **b** Functional information transfer through all 56 links established from eight peptide-GPCR pairs. Eight GPCRs at the *g1* position were coupled to secretion of the seven non-cognate peptides at the *p2* position. Heatmaps show the fluorescence/OD$_{600}$ value of *c2* after exposing *c1* to increasing doses of *p1*. Supplementary Figures 16 and 17 list full data sets and references heat maps. **c** Overview of the implemented communication topologies. Gray nodes: cells are able to process one input (expressing one GPCR) giving one output (secreting one peptide). Blue nodes: cells are able to process two inputs (OR gates, expressing two GPCRs) giving one output (secreting one peptide). Orange nodes: cells constitutively secrete the peptide for the next clockwise neighbor, and report on ring closure via a fluorescent read-out upon receiving a peptide signal from the counter-clockwise neighbor. Red nodes: cells are able to receive a signal and respond via a fluorescent read-out. **d** Ring topologies with an increasing number of members were established. An interrupted ring, with one member dropped out, was used as the control. Measurements were performed in triplicate, and error bars represent standard deviations. The fold-change in fluorescence between the full-ring and the interrupted ring is indicated for each topology. **e, f** A three-yeast bus topology (**e**) and a six-yeast branched tree-topology (**f**) were implemented (panel **c**). Fluorescence was measured after induction with all possible combinations of the three input peptides (zero, one, two, or three peptides). The numbers above the bars indicate the fold-change in fluorescence over the no-peptide induction value. Measurements were performed in triplicate, error bars represent SD

read-out (Supplementary Figure 18). We started with assembling a two- and a three-member ring (Fig. 3d and Supplementary Figure 19). An interrupted ring, with one member dropped out, was used as a control, and the results are reported as fold change in fluorescence between the full ring and the interrupted ring. We used colony PCR to assess the culture composition over time in the three-member ring. Due to differential growth behavior of individual strains (discussed in Supplementary Figure 20), we observed that single strains eventually took over the culture (Supplementary Figure 21). Than, in order to test for inherent scalability, we increased the number of members in the communication ring stepwise from three to six members (Fig. 3d and Supplementary Figure 19).

To test if we could achieve a different interconnected communication topology, we also implemented a branched tree topology using cells co-expressing two GPCRs and accordingly being able to process two inputs (dual-input nodes). Such topologies allow integration of multiple information inputs and report on the presence of at least one of these distributed inputs. We first tested functional signal flow through a three-yeast linear bus topology able to process two inputs (Fig. 3c, **topology 6**). We then added two branches upstream of the three-yeast bus and a side branch, eventually leading to a six-yeast tree with two dual-input nodes (Fig. 3c, **topology 7** and Supplementary Figures 22 and 23). To test functionality of communication, we started the information flow by adding the synthetic peptide ligand(s) recognized by the yeast cells starting each branch (we compared single, dual, and triple inputs) (Fig. 3e, f). Only the last yeast cell encoded a peptide-controlled fluorescent readout, enabling measurement once information traveled successfully through the topology by comparing the fold change in fluorescence compared with not adding starting peptide.

**Peptide-signal dependence enables interdependent communities**. Finally, to demonstrate the utility of our language, we made a synthetic, interdependent microbial community. We leveraged the orthogonal signal interfaces to render yeast cells mutually dependent based on peptide-signal control of essential gene expression. Engineered interdependence is of central importance for synthetic ecology, as it can be used to enforce the integrity of a synthetic community. Current approaches to engineer mutual dependence in synthetic communities rely on metabolite cross-feeding[46], which drastically limits the number of members that can be rapidly added to such a microbial community, and suffer from a dependence on cross-feeding metabolically expensive molecules needed at substantial molar concentrations. Our peptide–signal-based interdependence is conceptually different from cross-feeding metabolites as we use interfaces that are orthogonal to the cellular metabolism, which allow scaling the number of community members by peptide–GPCR gene swapping and which are sensitive enough to function at low nanomolar signal concentrations.

In order to engineer mutually dependent strain communities, we placed an essential gene under GPCR control (Fig. 4a). We chose SEC4 as the target essential gene due to its performance in a previous study[49]. We engineered an orthogonal Ste12* transcription factor and a set of tightly controlled orthogonal Ste12*-responsive promoters (OSR promoters), matching the dynamic range to the expected intracellular SEC4 levels (Supplementary Figure 24). We replaced the natural SEC4 promoter with one of the OSR promoters in strains expressing either the Bc.Ste2, Ca. Ste2, or Vp1.Ste2 receptors. As expected, the resulting strains were dependent on peptide for growth and showed peptide/growth $EC_{50}$ values in the nanomolar range, a concentration range achievable by secretion (Supplementary Figure 25). All strains were transformed with either of the two non-cognate constitutive peptide expression plasmids. The resulting six strains were used to assemble all three combinations of interdependent two-member links, and we verified their growth in strict mutual dependence over > 60 h ( > 15 doublings) (Supplementary Figure 26). The growth rate of the two-membered consortium was thereby dependent on the member identity, probably defined by the secreted amount of a given peptide and the dose-response characteristics of a given GPCR. We then scaled the interdependent community to three members and demonstrated stable mutually dependent growth of this three-member cycle over > 7 days ( > 50 doublings), while communities missing one essential member collapsed (Fig. 4b, c). We verified the presence of each strain and peptide over time (Fig. 4d and Supplementary Figure 27). Stable ratios of community members were not reached over the course of this experiment, suggesting that scaling in the number of members elicits more complex community behaviors. Mathematical modeling as well as experimental parameterization of peptide secretion rates and peptide-secretion-linked growth rates will be required to understand and harness these interesting dynamics. Once predictable, we envision that "peptide-signal interdependence" will allow fine-tuning the abundance of each strain in a consortium eventually allowing one to control abundance in space and time.

## Discussion

Inspired by the early impact of bacterial QS on our ability to engineer cell–cell communication and complex behavior, we repurposed fungal mating peptide-GPCR pairs into a signaling language with a scalable number of orthogonal interfaces. We demonstrate that the fungal pheromone response pathway naturally provides a large pool of unique signal and receiver interfaces that can be harnessed to build a modular, synthetic communication language. Importantly, these interfaces are readily accessible by genome mining, as both the peptides and the GPCRs are genetically encoded and can be implemented by simple gene cloning and expression.

Genome mining alone yields a high number of off-the-shelf orthogonal interfaces whose component diversity can potentially be further scaled and tuned by directed evolution to exploit the full information density of the 9–13 amino acid peptide ligands (sequence space $> 10^{14}$). Further, the language can be tuned by ligand recoding, as small changes in the sequence of a given peptide ligand alters the response behavior of a given GPCR. Importantly, changing the ligand sequence can be achieved by simple cloning and does not require receptor or metabolic engineering. In addition, peptides are technically ideal as a signal. Peptides are stable and rich in molecular information, and virtually any short peptide sequence is readily available through commercial solid-phase synthesis allowing for the rapid characterization and evolution of new peptide-sensing mating GPCRs.

Our peptide-GPCR language is modular and insulated, and thus likely portable to many other Ascomycete fungi, from which our component modules are derived. Furthermore, as has been done for mammalian GPCRs in yeast, this system is potentially portable to animal and plant cells. Its simplicity suggests that the system will be easy for other laboratories to adopt, scale, and customize, especially in the light of new tools for the rational tuning of GPCR-signaling in yeast[50]. Importantly, our language is compatible with existing and future synthetic biology tools for applications such as biosensing, biomanufacturing[51,52], or building living computers[37,53].

## Methods

**Strains**. Yeast strains and the plasmids contained are listed in Supplementary Table 4. All strains are directly derived from BY4741 (*MATa leu2Δ0 met15Δ0*

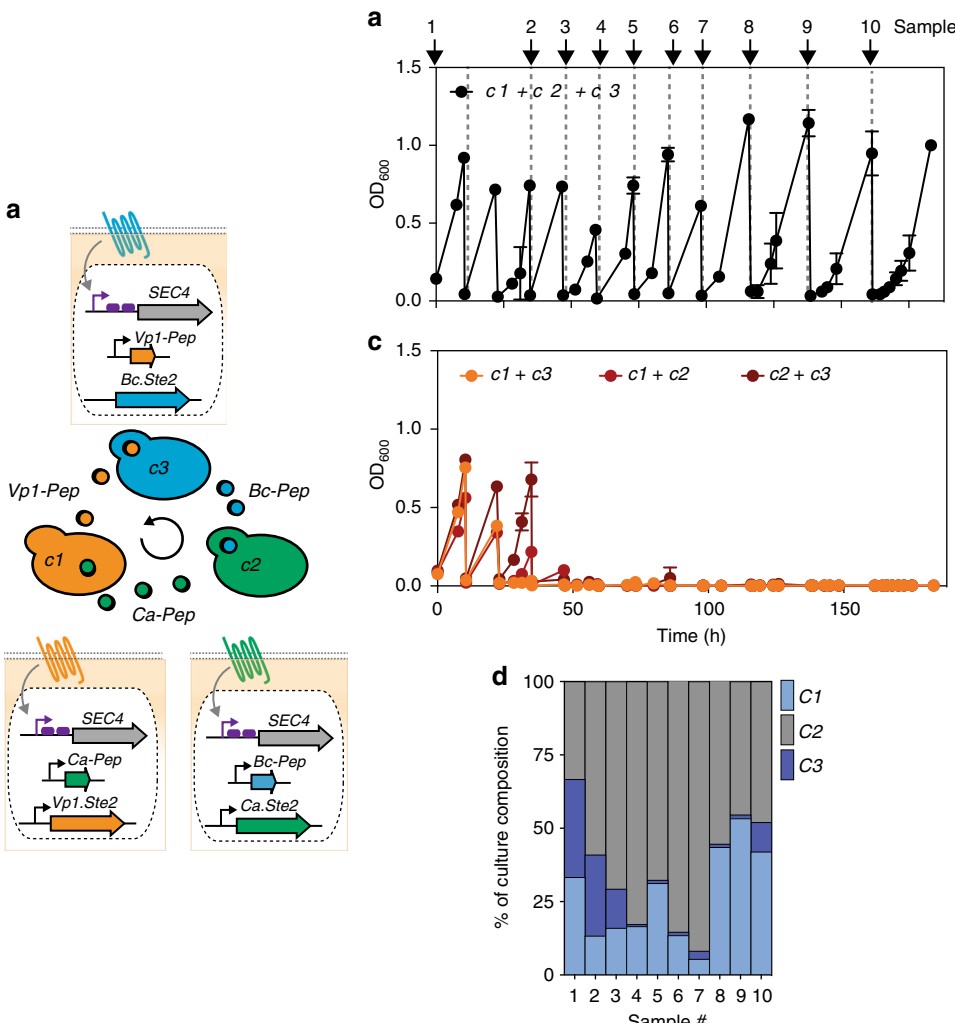

**Fig. 4** The synthetic communication language enables construction of an interdependent microbial community. **a** Illustration of the interdependent microbial communities mediated by the peptide-based synthetic communication language. Peptide-signal interdependence was achieved by placing an essential gene (*SEC4*) under GPCR control. In the featured three-yeast ring *c1*, *c2*, and *c3* secret the peptide needed for growth of the *cx-1* member of the ring. Peptides are secreted from the constitutive ADH1 promoter. **b**, **c** Growth of the three-membered interdependent microbial community over > 7 days. Communities with one essential member dropped out collapse after ~ 2 days (**c**). Three-membered communities were seeded in a 1:1:1 ratio, controls were seeded using the same cell numbers for each member as for the three-membered community. All experiments were run in triplicate and error bars represent the standard deviation. **d** The composition of the culture was tracked over time by taking samples from one of the triplicates at the indicated time points, plating the cells on media selective for each of the three component strains, and colony counting

ura3Δ0 his3Δ1) and BY4742 (*MATα leu2Δ0 lys2Δ0 ura3Δ0 his3Δ1*) by engineered deletion using CRISPR Cas9[54,55].

**Media**. Synthetic dropout media (SD) supplemented with appropriate amino acids; fully supplemented medium containing all amino acids plus uracil and adenine is referred to as synthetic complete (SC)[56]. Yeast strains were also cultured in YEPD medium[57,58]. *Escherichia coli* was grown in Luria Broth (LB) media. To select for *E. coli* plasmids with drug-resistant genes, carbenicillin (Sigma-Aldrich) or kanamycin (Sigma-Aldrich) were used at final concentrations of 75–200 μg/ml and 50 μg/ml, respectively. Agar was added to 2% for preparing solid yeast media.

**Materials**. Synthetic peptides (≥ 95% purity) were obtained from GenScript (Piscataway, NJ, USA). *S. cerevisiae* alpha-factor was obtained from Zymo Research (Irvine, CA, USA). Polymerases, restriction enzymes, and Gibson assembly mix were obtained from New England Biolabs (NEB) (Ipswich, MA, USA). Media components were obtained from BD Bioscience (Franklin Lakes, NJ, USA) and Sigma-Aldrich (St. Luis, MO, USA). Primers and synthetic DNA (gBlocks) were obtained from Integrated DNA Technologies (IDT, Coralville, Iowa, USA); Primers used in this study are listed in Supplementary Table 9. Plasmids were cloned and amplified in *E. coli* C3040 (NEB). Sterile, black, clear-bottom 96-well microtiter plates were obtained from Corning (Corning Inc.).

**Bioinformatic extraction of GPCR and peptide genes**. A database of fungal receptors were curated from the InterPro (IPR000366)[59] and PFAM (PF02116) families[60]. Sequence identifiers were standardized using the UniProt ID mapping tool (http://www.uniprot.org/uploadlists/). UniProt IDs were used to programmatically retrieve associated taxonomic information. Taxonomic information was used to filter out non-fungal sequences and fragments. The amino acid sequences of the corresponding peptide ligands were derived in a similar approach. Sequences were validated by multiple sequence alignment using Clustal Omega[61]. The amino acid sequences, as well as the percentage identity for all Ste2-like GPCRs and peptide precursors are listed in Supplementary Tables 1, 2, and 8).

**Code availability**. The custom code that was used for the programmatic retrieval of taxonomic information can be obtained from the authors upon request with no restrictions.

**Inference of the amino acid sequences of peptide ligands**. The amino acid sequences of the mature peptide ligands were either taken from literature (Supplementary Table 2) or predicted using the method reported by Martin et al.[62] In brief, mating pheromone precursor genes have a relatively conserved architecture. Genes encode for an N-terminal secretion signal (pre-sequence at the amino acid level), followed by repetitive sequences of the pro-peptide composed of non-

homologous pro-sequences, homologous sequences belonging to the presumptive signal peptide and protease processing sites. Based on this conserved arrangement, the actual sequence of the secreted peptide ligand can be predicted from the precursor sequence. Alignment with reported functional pheromone precursor sequences (from *S. cerevisiae* and *C. albicans*) facilitated annotation.

**Construction of GPCR expression vectors**. The GPCR expression vector is based on pRS416 (*URA3* selection marker, *CEN6/ARS4* origin of replication). All GPCRs were cloned under control of the constitutive *S. cerevisiae TDH3* promoter and terminated by the *S. cerevisiae STE2* terminator. Unique restriction sites (*Spe*I and *Xho*I) flanking the GPCR coding sequence were used to swap GPCR genes. Most GPCRs were codon-optimized for *S. cerevisiae*, DNA sequences were ordered as gBlocks, amplified with primers giving suitable homology overhangs, and inserted into the linearized acceptor vector by Gibson assembly. DNA sequences of all GPCR genes as well as the sequence of the full expression cassette (*TDH3*p-xy.Ste2-*STE2*t) integrated into the Δ*ste2* locus are listed in Supplementary Table 3. Amino acid sequences of all GPCRs are listed in Supplementary Table 8.

**Construction of peptide secretion vectors**. The peptide secretion vector is based on pRS423 (*HIS3* selection marker, 2μ origin of replication)[54]. The peptide coding sequence was designed based on the natural *S. cerevisiae* α-factor precursor as described previously[43]. In brief, to make a general secretion cassette, we amplified the *MFα1* gene with or without the Ste13 processing site (EAEA). The actual sequences for the peptide ligands were inserted via a unique restriction site (*Afl*II) after the pre- and pro-sequence, thus the peptide DNA sequence could be swapped by Gibson assembly[63] using peptide-encoding oligos codon-optimized for expression in yeast. The DNA and resulting protein sequences of all peptide precursor genes are listed in Supplementary Table 6. We used the constitutive *ADH1* promoter or the ligand-dependent *FUS1* and *FIG1* promoters to drive peptide expression. Promoters were amplified from *S. cerevisiae* genomic DNA.

**CRISPR-Cas9 system**. The Cas9 expression plasmid was constructed by amplifying the Cas9 gene with *TEF1* promoter and *CYC1* terminator from p414-*TEF1*p-Cas9-*CYC1*t[55] cloned into pAV115[64], using Gibson assembly[63]. For short genes, *MFALPHA1/2 and MFA1/2*, a single gRNA was cloned into a gRNA acceptor vector (pNA304) engineered from p426-*SNR52*p-gRNA.*CAN1*.Y-*SUP4*t[55] to substitute the existing *CAN1* gRNA with a *Not*I restriction site. gRNAs were cloned into the *Not*I sites using Gibson assembly[63]. Double gRNAs acceptor vector (pNA0308) engineered from pNA304 cloned with the gRNA expression cassette from p*RPR1*gRNAhandle*RPR1*t[65] with a *Hind*III site for gRNA integration. gRNAs were cloned into the *Not*I and *Hind*III sites using Gibson assembly[63]. For engineering yeast using the Cas9 system, cells were first transformed with the Cas9 expressing plasmid, followed by co-transformation of the gRNA carrying plasmid and a donor fragment. Clones were then verified using colony PCR with appropriate primers.

**Construction of core peptide-GPCR language acceptor strains**. Core *S. cerevisiae* strains yNA899 and yNA903 are derivatives of strain BY4741 (*MAT**a** leu2Δ0 met15Δ0 ura3Δ0 his3Δ1*) and BY4742 (*MATα lys2Δ0 leu2Δ0 ura3Δ0 his3Δ1*), respectively. They are deleted for both *S. cerevisiae* mating GPCR genes (*ste2* and *ste3*) and all mating pheromone-encoding genes (*mfa1, mfa2, mfα1, mfα2*) as well as for the genes *far1, sst2*, and *bar1*. All genes were deleted as open reading frame deletions using CRISPR/Cas9 as described below. In most cases, except for *MFA* genes, two gRNAs were designed for each gene to target sequences on the 5′ and 3′ end of the gene's open reading frame (all gRNA sequences are listed in Supplementary Table 7). Genes were deleted sequentially. After each round of gene deletion, strains were cured from the gRNA vector and directly used for deleting the next gene.

**Genomic integration of color read-outs and GPCR genes**. yNA899 was used to insert a *FUS1* and a *FIG1* promoter-driven yeast codon-optimized RFP (coRFP) into the HO locus. Using yeast Golden Gate (yGG)[64], we assembled a transcription unit of the appropriate promoter (*FUS1* or *FIG1*) with coRFP[66] coding sequence and a *CYC1* terminator into pAV10.HO5.loxP. Following yGG assembly and sequence verification, plasmid was digested with *Not*I restriction enzyme and transformed into yeast cells. Clones are then verified using colony PCR with appropriate primers. The resulting strain JTy014 was used for all GPCR characterizations by transforming it with the appropriate GPCR expression plasmids. GPCR genes were integrated into the Δ*ste2* locus of yNA899. The *TDH3*p-xySte2-*STE2*t expression cassette for Bc.Ste2, Sc.Ste2, Ca.Ste2, Kp.Ste2, Cl.Ste2, Cgu.Ste2, Hj.Ste2, So.Ste2, and Sj.Ste2 was used as repair fragment. The resulting generic locus sequence is listed in Supplementary Table 3.

**Construction of peptide-dependent yeast strains**. yNA899 was used as parent. First, expression cassettes for Bc.Ste2 and Ca.Ste2 were integrated into the Δ*ste2* locus as described above. We then replaced the DNA-binding domain of the pheromone-inducible transcription factor Ste12 (residues 1–215) with the zinc-finger-based DNA binding domain 43–8[67] (the resulting Ste12 variant is referred to

as orthogonal Ste12*, Supplementary Figure 24). We then replaced the natural *SEC4* promoter with differently designed synthetic orthogonal Ste12* responsive promoters (OSR promoters) and screened resulting strains for best performers (with regard to peptide-dependent growth). Resulting strains ySB270 (Ca.Ste2) and ySB188 (Vp1.Ste2) feature OSR4, strain ySB265 (Bc.Ste2) features OSR1. All genomic engineering steps were achieved using CRISPR-Cas9 and the guide RNAs are listed in Supplementary Table 7.

**GPCR on–off activity and dose–response assay**. GPCR activity and response to increasing the dosage of synthetic peptide ligand was measured in strain JTy014 using the genomically integrated *FUS1*-promoter-controlled coRFP as a fluorescent reporter. JTy014 strains carrying the appropriate GPCR expression plasmid were assayed in 96-well microtiter plates using 200 μl total volume, cultured at 30 °C and 800 rpm. Cells were seeded at an $A_{600}$ of 0.3 (note: all herein reported cell density values are based on $A_{600}$ measurements in 96-well plates of a 200 μl volume of cultures with a path length of ~0.3 cm performed in an Infinite M200 plate reader from Tecan) in SC media lacking uracil (selective component). All measurements were performed in triplicates. RFP fluorescence (excitation: 588 nm, emission: 620 nm) and culture turbidity ($A_{600}$) were measured after 8 h using an Infinite M200 plate reader (Tecan). Since the optical density values were outside the linear range of the photodetector, all optical density values were first corrected using the following formula to give true optical density values:

$$A_{true} = \frac{k \cdot A_{meas}}{A_{sat} - A_{meas}} \qquad (Eq.1)$$

, where $A_{meas}$ is the measured optical density, $A_{sat}$ is the saturation value of the photodetector, and $k$ is the true optical density at which the detector reaches half saturation of the measured optical density[32]. Dose–response was measured at different concentrations (11 fivefold dilutions in $H_2O$ starting at 40 μM peptide, $H_2O$ was used as no peptide control) of the appropriate synthetic peptide ligand. All fluorescence values were normalized by the $A_{600}$, and plotted against the log (10)-converted peptide concentrations. Data were fit to a four-parameter non-linear regression model using Prism (GraphPad) in order to extract GPCR-specific values for basal activation, maximal activation, $EC_{50}$, and the Hill coefficient. Fold-activation was calculated for each GPCR as the maximum $A_{600}$-normalized fluorescence of peptide-treated cells divided by the $A_{600}$ normalized fluorescence value of water-treated cells.

**GPCR orthogonality assay using synthetic peptides**. GPCR activation was individually measured in 96-well microtiter plates in triplicate using each of the synthetic peptides (10 μM). Cells were seeded at an $A_{600}$ of 0.3 in 200 μl total volume in 96-well microtiter plates, cultured at 30 °C and 800 rpm. Endpoint measurements were taken after 12 h, as described above. Percent receptor activation was calculated by setting the $A_{600}$-normalized fluorescence value of the maximum activation of each GPCR (not necessarily its cognate ligand) to 100% and the value of water-treated cells to 0%, with any negative values set to 0%.

**Peptide secretion fluorescent halo assay**. JTy014 was transformed with the appropriate GPCR expression plasmid, and resulting strains were used as sensing strains. yNA899 was transformed with the appropriate peptide secretion plasmids and used as secreting strains. Sensing strains for all 16 peptides were individually spread on SC plates. Briefly, 0.5% agar was melted and cooled down to 48 °C, cells are added to an aliquot of agar in a 1:40 ratio (100 μL of cells into 4 mL of agar for a 100 mm petri dish and 200 μL of cells into 8 mL of agar for a Nunc Omnitray), mixed well, and poured on top of a plate containing solidified medium. A 10 μL dot of each of the secreting strains was spotted on each of the sensing strain plates. Plates were incubated at 30 °C for 24–48 h and imaged using a BioRad Chemidoc instrument and proper setting to visualized RFP signal (light source: Green Epi illumination and 695/55 filter).

**Peptide secretion liquid culture assay**. We examined peptide secretion in liquid culture by co-culturing a secreting and a sensing strain (expressing the cognate GPCR) and measuring fluorescence of the induced sensing strain. Peptide secretion was under control of the constitutive *ADH1* promoter. Secretion strains for each peptide were constructed by transforming yNA899 with the appropriate peptide expression construct (pRS423-*ADH1*p-xy.Peptide) along with an empty pRS416 plasmid. Sensor strains were constructed by transforming JTy014 with the appropriate GPCR expression construct (pRS416-*TDH3*p-xy.Ste2) along with an empty pRS423 plasmid. Matching the auxotrophic markers of the secretion and sensor strains allowed for robust co-culturing. Secreting and sensing strains were seeded in a 1:1 ratio each at an $A_{600}$ of 0.15, and $A_{600}$ and red fluorescence were measured after 12 h. Experiments were run in triplicate. An unpaired *t* test was performed for each peptide with an alpha value=0.05 to determine if differences in secretion between constructs containing or not containing the Ste13 processing site were significant. A single asterisk indicates a *P*-value < 0.05; a double asterisk indicates a *P*-value < 0.01.

**Secretion orthogonality assay.** The same sensing and secreting strains as described for the "Peptide secretion liquid culture assay" (above) were used to confirm orthogonality of secreted peptide in co-culture. Only the constructs that retained the Ste13 processing site were used. To determine orthogonality, each of the 16 constructed secretion strains were co-cultured 1:1 each at an $A_{600}$ of 0.15 with the corresponding sensor strains to test for GPCR activation by non-cognate peptide, and $A_{600}$ and red fluorescence were measured after 14 h. Experiments were run in triplicate. Percent activation of the sensor strain was normalized by setting the maximum observed activation of the sensor strain (not necessarily by the cognate ligand) to 100%, and setting the basal fluorescence from co-culturing each sensor strain with a non-secreting strain to 0% activation, with any negative values set to 0%.

**Transfer functions through minimal communication units.** yNA899 with the appropriate GPCR integrated into the Ste2 locus using the CRISPR system (described above) was transformed with the appropriate peptide secretion plasmid (pRS423-*FIG1*p-xy peptide retaining the Ste13 processing site), and the resulting strains were used as cell 1 (*c1*, sender). JTy014 was transformed with the appropriate GPCR expression plasmid (pRS416-*TDH3*p-xy.Ste2) and used as cell 2 (*c2*, reporter). As *c1* and *c2* didn't have the same auxotrophic markers, validated strains were grown overnight in selective media and then seeded at a 1:1 ratio each at an $A_{600}$ of 0.15 in SC media. Cells were cultured in a total volume of 200 µl in 96-well microtiter plates, and *c1* was induced with the appropriate synthetic peptide at 2.5 nM, 50 nM, and 1000 nM, using water as the 0 nM control. Red fluorescence and $A_{600}$ were measured after 12 h. As a control, *c2* was co-cultured with a non-secreting strain carrying an empty pRS423 plasmid and induced with the appropriate synthetic peptide at the concentrations listed above.

**Multi-yeast paracrine ring assay.** Communication loops were designed so that a single fluorescent measurement would indicate signal propagation through the full ring topology. An initiator strain was constructed by integrating the Ca.Ste2 into JTy014 and transforming it with a constitutive Kp peptide secretion plasmid (pRS423-*ADH1*p-Kp.Peptide). Linker strains from the transfer functions experiment (without a fluorescent readout) were used to complete each communication ring. Communication rings were seeded in triplicate at equal ratios ($A_{600} = 0.02$ each) in 10 mL selective 2x SC–His medium and incubated at 30 °C with 250RPM shaking for 36 h. In total, 200 µL samples were taken for a fluorescent measurement of red fluorescence (588 nm/620 nm excitation/emission) in technical triplicate in a 96-well black clear-bottom plate and normalized by $A_{600}$. To demonstrate that communication is contingent on a complete ring topology, a control with the first linker yeast strain in each ring dropped out was performed in parallel. The panels compare the normalized red fluorescent signal for each ring to the dropout control, with the fold-change induction of the completed ring indicated.

**Tree topology assay.** Bus and tree topologies were designed so that a single fluorescent measurement would indicate signal propagation through the full topology. To enable branched topologies with two-input nodes, an additional orthogonal GPCR was integrated into the *STE3* locus using the CRISPR–Cas9 system described above (strains ySB315 and ySB316, Supplementary Table 4). Single and dual dose-response characteristics of ySB315 and ySB316 confirmed the ability to activate either or both co-expressed GPCRs (Supplementary Figure 7). ySB315 and ySB316 were then transformed with the appropriate peptide secretion plasmids and combined with linker strains validated from the transfer functions experiment and ySB98 transformed with an empty pRS423 plasmid as a fluorescent readout of communication. Communication topologies were seeded at equal ratios ($A_{600} = 0.02$ each) in 10 mL selective 2x SC–His medium and incubated at 30 °C with 250RPM shaking for 16 h. In total, 200 µL samples were taken for a fluorescent measurement of red fluorescence (588 nm/620 nm excitation/emission) in technical triplicate in a 96-well black clear-bottom plate and normalized by $A_{600}$. To demonstrate that dual-input nodes may be activated by either one or two-input peptides, different combinations of the input peptides were added at 1 µM each (see Supplementary Figure 23 for key to Fig. 3e, f). Fold change compared with no added peptide is indicated.

**Flow cytometry.** Cells were seeded at an $A_{600}$ of 0.3. Cells were exposed to the indicated peptide concentrations and cultured for 12 h in 96-well microtiter plates in a total volume of 200 µl at 30 °C and 800 rpm shaking. For each sample, 50,000 cells were analyzed using a BD LSRII flow cytometer (excitation: 594 nm, emission: 620 nm). The fluorescence values were normalized by the forward scatter of each event to account for different cell size using FlowJo Software.

**Peptide-dependent growth assay.** Strains ySB270, ySB265, and ySB188 were maintained on SD agar plates supplemented with 1 µM of Ca, Bc, or Vp1 peptides. For assaying their peptide-dependent growth response, strains were cultured overnight in the presence of 100 nM peptide in SC–His. Cells were washed five times with one volume of water. Cells were then seeded in 200 µl SC (no selection) at an $A_{600}$ of 0.06 and cultured at 30 °C and 800 rpm shaking. Cells were exposed to different concentrations of peptide (seven 10-fold dilutions starting from 1 µM, water was used for the "no-peptide" control). $A_{600}$ was determined at various time

points over the course of 24 h. The 24 h-data points were plotted against the $\log_{10}$ of the peptide concentrations. Data were fit to a four-parameter non-linear regression model using Prism (GraphPad) to extract values for peptide/growth $EC_{50}$. For *dot assays*, serial 10-fold dilutions of overnight cultures of ySB270 and ySB265 were spotted on SD agar plates supplemented with or without 1 µM peptide and incubated at 30 °C for 48 h.

**Two-Yeast and Three-Yeast interdependent co-culturing.** Strains ySB270, ySB265, and ySB188 were transformed with the appropriate peptide secretion vectors (Bc, Ca, or Vp1) featuring peptide expression under the constitutive *ADH1* promoter. For assaying two-yeast interdependence, the resulting peptide-secreting strains (treated with peptide and washed as described above) were seeded in the appropriate combination in a 1:1 ratio in 200 µl SC–His at an $A_{600}$ of 0.06 (0.03 each) and cultured at 30 °C and 800 rpm shaking. The same cell number of single strains was seeded alone and cultured in parallel as control. $A_{600}$ measurements were taken at the indicated time points and cultures were diluted into fresh media when the culture reached an $A_{600}$ of 0.8 -1. For assaying three-yeast interdependence, the appropriate peptide secreting strains (*c1*, *c2*, and *c3*) were inoculated in a ratio of 1:1:1 in 200 µl SC–His media at an $A_{600}$ of 0.06 (0.02 each) in a 96-well plate cultured at 30 °C and 800 rpm shaking. Experiments were run in triplicate. All three combinations of controls lacking one essential member (*c1* omitted, *c2* omitted, *c3* omitted) were run in parallel. $A_{600}$ measurements were taken at the indicated time points, and cultures were diluted 1:20 into fresh media approximately every 12 h. After 115 h, the dilution rate was reduced to 1:20 every 24 h. The total run time was 183 h (~ 7.5 d). Samples were taken before every dilution. Samples were used to determine the co-culture composition and the peptide concentration. Deconvolution of strain identity: aliquots of the culture were plated on three different plate types, YPD containing either 1 µM Bc, Ca, or Vp1 synthetic peptide. Each strain can only grow on plates containing its cognate peptide ligand. The co-culture composition was than determined by colony counting. Peptide concentration: We used JTy014 transformed with the appropriate GPCR as peptide sensor. The linear range of the GPCR dose response was used for peptide quantification.

## Data availability
The authors declare that all the data supporting the findings of this study are available within the paper and its supplementary information files or from the authors upon reasonable request.

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

## Acknowledgements

We thank N. Michael, R. McBee, J.M. de Flores Quijano, and B. Oh for their experimental contributions. This research was partly funded by DARPA award HR0011-15-2-0032 and NIH award 5R01AI110794. S.B. was supported by a Simons Junior fellow award from the Simons Foundation. M.J. and J.B. were supported by NSF Graduate Research Fellowships (DGE 16–44869). M.S. is supported by T32 GM066704 (Bach). Part of the research reported in this publication was performed in the CCTI Flow Cytometry Core at Columbia Medical Campus, supported in part by the NIH Office of the Director, under awards S10RR027050. The content is solely the responsibility of the authors and does not necessarily represent the official views of the NIH.

## Author contributions

V.W.C, M.J., and S.B. conceived and conceptualized the idea that the fungal peptide-GPCR pairs could be a scalable language and provided preliminary data for the scalable language. M.J. performed programmatic retrieval of GPCR and peptide genes and extracted candidate signal peptide sequences. M.J., J.B., and S.B. cloned and characterized the GPCRs. N.A. and M.S. designed the core read-out strain. N.A. constructed the core read-out strains. N.A., J.T., J.B., and S.B. constructed strain derivatives. N.A. and J.T. cloned the peptide expression cassettes. N.A., J.B., and J.T. performed peptide secretion experiments. N.A. and J.T. performed the Halo assay. J.B. performed the multi-yeast communication experiments. S.B. performed peptide-signal dependent co-culture experiments. S.B., J.B., and N.A. designed the experiments and analyzed the data. S.B. wrote the paper with the help of all authors. J.D.B. and V.W.C. supervised the work.

## Additional information

**Competing interests:** A provisional patent application (62/516,383) was filed June 7, 2017 naming V.W.C. as inventor and assigned to The Trustees of Columbia University in the City of New York. J.D.B. is a founder and Director of the following: Neochromosome, Inc., the Center of Excellence for Engineering Biology, and CDI Labs, Inc. and serves on the Scientific Advisory Board of the following: Modern Meadow, Inc., Recombinetics, Inc., and Sample6, Inc. The remaining authors declare no competing interests.

