## [Peer Review File · Nature Communications]

This manuscript has been previously reviewed at another journal that is not operating a transparent peer review scheme. This document only contains reviewer comments and rebuttal letters for versions considered at Nature Communications . Mentions of the other journal have been redacted.

Reviewer #1 (Remarks to the Author):

In this revision, the authors have properly addressed all my previous concerns. The manuscript is ready to be published in Nature Communication.

No changes requested.

Reviewer #2 (Remarks to the Author):

In the revision, the authors have clarified most of the technical and conceptual points I raised in my initial review, where I noted the usefulness of having these components and questioned the claimed significance (as initially stated).

To address this point, the authors went into more details on why they believed their mined components are superior than QS components.

I do not disagree with the power of genome mining in obtaining new components. However, this is an approach that has been demonstrated in previous studies in different contexts and thus, it's not the innovation of the present study.

I also do not disagree with having a pool of such components. What I was challenging is the strong claim that they're superior to components in different organisms. For instance, many criticisms on the limitation of bacterial QS systems are in fact based on speculation instead of being supported by concrete studies. For instance, the authors state:

“However, the scalability of QS into many independent channels is limited by the low information content that can be encoded in AHL signaling molecules, since these molecules are structurally and chemically simple and the receptors are known to be promiscuous.”

This is clearly speculation. Moreover, quite a few, but not all, components in the cited examples show a lower degree of promiscuity than the data presented in the manuscript under consideration. That is, even though the same word “promiscuous” is used, it implies different criteria in different studies. And many of the QS systems have higher dynamic ranges than the components described in the current study. The seemingly shortage of QS systems does not necessarily imply that QS systems are worse; it could simply mean that people haven't looked/mined hard enough.

As detailed in my previous comments, I am not particularly concerned with the presence of crosstalk in the existing systems or in the components described in this paper. The critical question is whether the target system design can tolerate the crosstalk, which is a question that the current study is not addressing. This is fine but making sweeping claims based on speculation is unnecessary and unhelpful.

I do not believe the authors need this claim for the study to be valuable to the community. Instead, I would suggest the authors to refine their arguments to make them more rigorous.

In order to accommodate the reviewers concern we edited the introduction to be less

comparative (peptide/GPCR vs. QS) but rather simply stating the current status of the QS language:

“The major class of QS is based on diffusible acyl-homoserine lactone (AHL) signaling molecules generated by AHL synthases and AHL receptors that function as transcription factors, regulating gene expression in response to AHL signals. Currently, only four AHL synthase/receptor pairs are available for synthetic communication, with three pairs successfully used together¹⁹. Scaling the QS components to make new orthogonal communication interfaces is challenged by the fact that many of the known receptors exhibit crosstalk^{21, 22}. While it is possible to eliminate crosstalk by receptor evolution²³, scaling the number of unique AHL ligand/receptor pairs by laboratory evolution requires the concerted engineering of AHL biosynthesis and receptor specificity.

Communication has also been engineered using autoinducer peptides (AIP)²⁴ and autoinducer molecules (AI-2)²⁵ from Gram-positive bacteria, however scaling is also challenged by the interdependence of multiple required signaling components. Autoinducer peptides are a class of post-translationally modified peptides sensed by a membrane-bound two-component system²⁶. AI-2 is a family of 2-methyl-2,3,3,4-tetrahydroxytetrahydrofuran or furanosyl borate diester isomers, synthesized by LuxS from S-ribosylhomocysteine followed by cyclization to a range of AI-2 isoforms^{27, 28}, and recognized by the transcriptional regulator LsrR²⁹. It was shown that the response characteristics and the promoter specificity of LsrR can be engineered^{30, 31} and that cell-cell communication can be tuned by using various AI-2 analogues²⁵. However, the complexity of signal biosynthesis and reliance on specific transporters for signal import and export²⁹ complicates the potential scalability of these systems in terms of available unique communication interfaces.”

Reviewer #3 (Remarks to the Author):

The authors have provided an adequate response to this reviewer's concerns. In particular, they have described that the success rate for the development of GPCR/peptide pairs is sufficiently positive so that hundreds of orthogonal channels would be found by others using the technique. I would like to see elements of this discussion reinforced. I'm not sure if a few sentences are best on P6 or later in the conclusions, but reinforcing the hit rate and the limited (not global) methodology would be good.

We incorporated this discussion into the last paragraph of the introduction:

“Our language acquisition pipeline showed a hit rate of 71%. Out of 45 tested GPCRs, 32 were functionally expressed and activated by a peptide ligand that was correctly inferred from its genomic locus architecture. Of these, 50% were highly orthogonal, yielding 17 unique communication channels without engineering. Importantly, our set included peptide/GPCR pairs derived from a wide range of species from the whole Ascomycete phylum. As such, we expect that many (likely hundreds or even thousands) additional orthogonal channels are available for extraction using the workflow described herein.”